# Educational Gradients in Drinking Amount and Heavy Episodic Drinking among Working-Age Men and Women in Spain

**DOI:** 10.3390/ijerph19074371

**Published:** 2022-04-05

**Authors:** Marta Donat, Gregorio Barrio, Juan-Miguel Guerras, Lidia Herrero, José Pulido, María-José Belza, Enrique Regidor

**Affiliations:** 1Escuela Nacional de Sanidad, Instituto de Salud Carlos III, 28029 Madrid, Spain; ma.donat@isciii.es (M.D.); gbarrio@isciii.es (G.B.); mbelza@isciii.es (M.-J.B.); 2CIBER Epidemiología y Salud Pública (CIBERESP), 28029 Madrid, Spain; josepuli@ucm.es (J.P.); enriqueregidor@hotmail.com (E.R.); 3Centro Nacional de Epidemiología, Instituto de Salud Carlos III, 28029 Madrid, Spain; 4Instituto Valenciano de Estadística, 46004 Valencia, Spain; lidia.herrero.huertas@gmail.com; 5Departamento de Salud Pública y Materno-Infantil, Universidad Complutense de Madrid, 28040 Madrid, Spain; 6Instituto de Investigación Sanitaria del Hospital Clínico San Carlos (IdISSC), 28040 Madrid, Spain

**Keywords:** alcohol use, average drinking amount, heavy episodic drinking, educational gradient, sex, Spain

## Abstract

Alcohol-related harm decreases as socioeconomic position increases, although sometimes the opposite happens with alcohol intake. The objective was to know the educational gradient in monthly measures of drinking amount and heavy episodic drinking (HED) among people aged 25–64 years in Spain from 1997–2017. Such gradient was characterized with the relative percent change (PC) in drinking measures per year of education from generalized linear regression models after adjusting for age, year, region, marital status and immigration status. Among men, the PCs were significantly positive (*p* < 0.05) for prevalence of <21 g alcohol/day (2.9%) and 1–3 HED days (1.4%), and they were negative for prevalences of 21–40 g/day (−1.1%), >40 g/day (−6.0%) and ≥4 HED days (−3.2%), while among women they ranged from 3.6% to 5.7%. The gradient in prevalences of >40 g/day (men) and >20 g/day (women) was greatly attenuated after additionally adjusting for HED, while that of ≥4 HED days was only slightly attenuated after additionally adjusting for drinking amount. Among women, the gradients, especially in HED measures, seem steeper in 2009–2017 than in 1997–2007. Educational inequality remained after additional adjustment for income and occupation, although it decreased among women. These results can guide preventive interventions and help explain socioeconomic inequalities in alcohol-related harm.

## 1. Introduction

Preventing alcohol use is essential to reduce the global burden of disease. The amount of alcohol intake and drinking patterns throughout life are the main determinants of alcohol-related health harm [1,2]. Knowing the distribution of alcohol drinking behavior according to socioeconomic position (SEP) is relevant to implement more effective and efficient preventive interventions [3], as well as to better characterize and interpret the so-called alcohol-harm paradox. In this regard, there is consistent evidence that alcohol-related harm, including mortality and morbidity, is significantly greater in low SEPs than in high SEPs [4,5,6,7,8,9,10]. Alcohol-attributable mortality is two to five times greater in the lowest SEP compared to the highest SEP (negative or inverse SEP gradient), setting a relative inequality greater than all-cause mortality [6,7]. This happens in both men and women and in practically all countries [7,9,11,12]. Surprisingly, a lower drinking prevalence or even mean amount of alcohol intake has often been found in low SEPs compared to high SEPs (positive SEP gradient) [8,13,14,15,16], a discrepancy known as the alcohol-harm paradox. This paradox is supported by the results of some research showing that the inverse SEP gradient in alcohol-related harm remains barely changing [4,15] or only attenuating [8,16,17,18] when adjusting for measures of frequency or amount of alcohol intake. Consequently, it has been thought that other factors, in addition to alcohol intake, should contribute to configure the inverse gradient in alcohol-related harm [4,12,15,18,19,20,21,22]. Among them would be reverse causation and differences according to SEP in underreporting of alcohol intake, access to services for alcohol problems, susceptibility to harmful alcohol effects, or risk factors that can act synergistically with alcohol intake (i.e., obesity, poor diet/exercise or tobacco smoking) [8,15,16,20,23].

However, the paradox can also be qualified. Although the SEP gradient is usually positive for drinking prevalence in both sexes in all countries, this does not happen with all drinking measures. Thus, the SEP gradient in prevalence of heavy average drinking (HAD), defined as exceeding a certain daily or weekly amount of alcohol intake, is usually negative among men in most countries, but positive among women in many countries, especially in low- and middle-income countries [14,24,25,26,27,28]. The inverse SEP gradient in HAD prevalence among men runs in the same direction as the gradient in alcohol-related harm. This is to be expected because dose−response curves between an individual amount of alcohol intake and many health conditions show that the risk of harm to health increases almost exponentially after a certain threshold of average daily amount of alcohol intake (i.e., 20–40 g), both in men and women [1,2]. In contrast, there is generally no significant increased risk of harm for lower amounts of alcohol intake. Therefore, it should not be surprising if positive gradients are found for prevalence of a low drinking amount or any drinking amount (since the drinking amount is low in most drinkers), nor if they are found for average drinking amount per capita (in research that allocates the same amount to all individuals in each socioeconomic group, masking the actual intake of each particular individual). What is really paradoxical is to find, as happens in many countries, a positive SEP gradient in HAD prevalence among women [14,25,26,27,28].

Regarding heavy episodic drinking (HED) the information is scarcer and generally refers to HED prevalence, having found an SEP gradient usually negative among men and negative in some countries and positive in others among women [13,14,20,26,29]. 

Exploring the heterogeneity in the SEP gradients by sex in Spain is interesting because it is an upper-middle income country where the incorporation of women into the workforce and into alcohol intake has accelerated throughout the 21st century. Regarding the drinking measures considered, although there is evidence that they can strongly influence the direction and magnitude of the SEP gradients, there is little research explicitly focused on this aspect. Very little is known, for example, about the change in SEP gradients with increasing HED frequency (most previous research having focused on the prevalence of any HED episode). There is also almost no information on the threshold in average drinking amount from which the positive SEP gradient is reversed among men (something that could be related to compliance with public health recommendations on harmful drinking by different socioeconomic groups). On the other hand, there are studies that estimate the inequality in drinking measures among drinkers [30,31], without warning readers that the results may be very different from those found among the population. In fact, in the USA in 2015, the educational gradient in the monthly prevalence of HED was positive among the population and negative among current drinkers [29].

A relevant topic that is not fully clarified is the relative contribution of average drinking amount (i.e., HAD) and HED to socioeconomic inequalities in alcohol-related harm. Elucidating it is not easy because drinking amount depends in part on the HED frequency (i.e., high HED frequencies usually lead to high drinking amounts) [32], and both drinking dimensions are usually associated with SEP [18]. Some evidence from cohort studies conducted in Anglo-Saxon and Northern European countries suggests that HED may be as or more important than HAD in explaining SEP inequalities in alcohol-related harm [17,18,23,33]. However, this may not necessarily be the case in countries, such as Spain, where it is likely that a significant portion of alcohol intake still comes from non-HED drinking patterns, such as the daily intake of wine or beer linked to meals. Due to its design, this research does not allow obtaining direct evidence on the relative contribution of both drinking dimensions on socioeconomic inequalities in alcohol-related harm, but it does allow estimating to what extent the contribution of each of these two dimensions is mediated by the other dimension. Thus, if the SEP gradient in prevalence of a high HED frequency were greatly attenuated after adjusting for drinking amount, this would mean that most of the contribution of HED frequency would be mediated by average drinking amount.

Knowing the temporal evolution of the SEP gradients in drinking measures is relevant in order to design and evaluate preventive interventions for alcohol intake and also to test the theory of diffusion of innovations. According to this theory, the early adopters of some new behaviors (e.g., tobacco smoking or HED) are high-SEP individuals or societies, and then such behaviors move to other socioeconomic positions [13,34,35,36]. Although considerable research has been published on the evolution of inequalities in alcohol-related harm, particularly mortality, the evolution of inequality in drinking behaviors has been less frequently reported. In general, previous findings suggest that in recent years, the relative SEP inequality in alcohol-related mortality (favoring higher SEPs) has increased [37,38,39,40,41], although not in all countries [42]. Therefore, results in the same direction could be expected with regard to SEP inequalities in the highest-risk drinking behavior, i.e., HAD or HED prevalence. However, the evolution of inequalities in such behaviors varies widely by sex and country [14,43,44,45,46,47]. In any case, if reality conforms to the innovation diffusion theory, the magnitude of the positive SEP gradient in drinking measures (higher values in higher SEP) among women should have decreased in recent years in Spain.

It is proposed to test the following research hypotheses on the educational gradient in drinking measures in the Spanish working-age population from 1997–2017: (1) The gradient will vary widely according to sex and the specific drinking measure. The gradient is expected to be positive for low drinking amounts and HED frequencies and to reverse from a certain threshold and end up being clearly negative for high drinking amounts and HED frequencies among men. However, the gradient is expected to be positive for all drinking measures, among women, although its magnitude will decrease as the drinking amount and HED frequency increase. (2) The educational gradient in drinking amount contributes to explain an important part of the gradient in HED frequency measures and vice versa. (3) The gender gap of the educational gradient in the highest-risk drinking behaviors has narrowed in recent years; more specifically, the negative gradient among men has become steeper and the positive gradient among women has become less steep.

The specific objectives of this work are to know the educational inequality in various monthly measures of drinking amount and heavy episodic drinking (HED) by sex and period among people aged 25–64 years in Spain from 1997–2017 and the effect of HED frequency on educational inequality in drinking amount and vice versa.

## 2. Materials and Methods

### 2.1. Study Population and Variables

Study participants were extracted from ten biennial editions of national household survey on alcohol and drug use in Spain (EDADES) conducted between 1997 and 2017 [48]. This survey included people aged 15–64 years living in family dwellings. The edition sample size ranged from 12,034–28,324. Collective dwellings (hotels, hostels, barracks, convents, prisons, student or elderly homes, etc.), precarious accommodations (shacks, tents, etc.) and homeless individuals were excluded from the sampling frame. A three-stage random sampling design (census section, household, individual) was used. Census sections and subjects within the households were selected by random probabilistic methods. Households within the census sections were selected by systematic random routes. The sample was stratified by age (15–34/35–64) and living region (19 categories). People aged 15–34 and small communities were oversampled. The response rate was about 50%, and the main causes of non-response were not opening the door or preventing someone in the home from being interviewed and absence of all household members or the selected person. Before classifying a household/person as absent and selecting another household, the fieldworker should visit the household initially selected at least three times on different days and times.

Data on substance use were collected through a self-administered questionnaire at home (paper-and-pen) and other data through face-to-face interview. Regarding alcohol in the last 30 days, people were asked about: (a) frequency (number of days) and quantity (number of standard drinks/day) of intake of major alcoholic beverages (wine, beer/cider, vermouth/alcoholic aperitifs, fruit liquors, strong liquors and cocktail spirits), and (b) frequency (number of days) of HED (intake of 5+ standard drinks in a two-hour interval; 4+ drinks in women from 2009 onward). Regarding SEP, people were asked to state their highest education category achieved (6–16 categories), the occupation of those who were employed during the interview week or had worked before (10–19 categories), and the total monthly household income and number of household members (only for 2007–2017). Likewise, they were asked about age, sex, living region, country of birth and marital status. Details on the survey methodology can be consulted in [48].

### 2.2. Statistical Analysis

Various outcomes on the amount of alcohol intake and HED frequency during the last 30 days were analyzed. Regarding the first, the outcomes were prevalence of any amount (drinking prevalence), several daily amount interval prevalences, including that of heavy average drinking (HAD), defined as the proportion of participants with an intake of >40 g/day among men and >20 g/day among women and mean amount per participant in g/day (mean daily drinking amount). The amount of alcohol intake for each individual was calculated by adding the alcohol from the six beverage types mentioned above, following the quantity−frequency approach. For each beverage, the daily amount of alcohol in grams was obtained by multiplying the number of days of intake in the last 30 days by the mean number of standard drinks/day and the grams of alcohol in a standard drink of such beverage and dividing the product by 30. Given the scarcity of updated empirical data, the alcoholic content in grams in the standard drink of each beverage was taken from previous studies [48,49], specifically 11.4 (wine), 10.9 (beer/cider), 11.8 (vermouth/alcoholic aperitifs), 10.0 (fruit liquors), 16.6 (strong liquors) and 16.6 (cocktails spirits), figures that in turn were estimated by multiplying the volume of a standard drink in milliliters (from clinical or public health guidelines) by the percent of alcohol-by-volume of such a beverage used by the Spanish Tax Agency and by the alcohol density (0.79). The amount of alcohol intake could not be obtained from EDADES 2013 due to flaws in the questionnaire questions. Regarding HED, outcomes were prevalence of any HED episode (HED prevalence), several interval prevalences of number of HED days and mean HED days.

The main SEP measure was the highest educational level attained, classified into four categories (primary, secondary 1st stage, secondary 2nd stage and university), corresponding, respectively, to levels 0–1, 2, 3–4 and ≥5 of the 2011 International Standard Classification of Education [50]. Alternatively, the education categories in the questionnaire were transformed into number of years of education. Secondarily, other SEP measures such as occupational class and income level were also considered. Four occupational classes were derived from the Erikson−Goldthorpe−Portocarrero social class scheme: managers/professionals, intermediate professionals/non-manual workers, skilled manual workers and unskilled manual workers. The individual income level was obtained by dividing the annual household income by the number of household members and classifying the result into four categories. The education level has the advantage, compared to occupational class and income level, of being more stable over time and less affected by reverse causation [51].

Survey counts were weighted to account for strata imbalance compared to the universe according to age and region. The analysis of inequalities focused on the population aged 25–64 years. The proportion of missing values for the amount of alcohol intake, HED frequency, education, occupation and income in this population was 11.0%, 1.7%, 0.9%, 1.8% and 26.0%, respectively, and only cases with known values for amount of alcohol intake and the SEP indicator were selected. The effective sample size in the bivariate analysis by education level was 67,590 men and 69,662 women and that in the stratified analysis by education, sex, age group and period is shown in Appendix A.

Regarding educational inequality, age-standardized drinking measures by education level were first calculated using the 2013 European Standard Population weights, as well as the statistical significance of their linear trends across education levels. Subsequently, two alternative indicators of relative educational inequality of drinking measures were estimated from negative binomial regression models with robust variance and log link in the framework of generalized linear models, adjusting for simple age, simple calendar year, region (18 categories), marital status (four categories) and immigration status (born in Spain or abroad), all of them entered as dummy variables. The specific indicators were: (a) the ratios of the drinking measures in each education level to university level by introducing the education level as a dummy variable; (b) the educational gradient or relative percent change in drinking measures per year of education (PC_ye_) by introducing the number of education years as a quantitative variable and transforming its beta regression coefficient into a more easily interpretable PC_ye_ [PC_ye_=100 (e^β^−1)] (model I).

To clarify the association of education with drinking amount and HED frequency, new estimators of educational inequality were obtained by adding to model I the HED frequency in days as a categorical adjustment covariate, when the outcome was a drinking amount measure, and the drinking amount in grams of alcohol/day, when the outcome was an HED frequency measure (model II). The 95% confidence intervals (95% CI) of age-standardized drinking measures and inequality estimators were obtained. All reported P-values are two-sided, with *p* < 0.05 considered statistically significant. All analyses were stratified by sex, although some indicators of educational inequality in the main drinking measures were also obtained for the four subgroups resulting from combining sex and period (1997–2007 and 2009–2017). Finally, complementary analyses were carried out, estimating the inequality in the main drinking measures according to occupational class or income level. Specifically, from multivariate models similar to those mentioned above (model I), the ratios were estimated in each occupational or income level up to the highest level, as well as the income gradient or percent change in drinking measures per thousand euros of annual income. All analyses were performed with IBM SPSS Statistics, Version 26.0 (IBM Corp.: Armonk, NY, USA). The Carlos III Health Institute review board approved the study proposal. Ethics committee approval was not required.

## 3. Results

### 3.1. Crude Drinking Measures and General Characteristics by Education Level

The crude measures of alcohol intake during the last 30 days by sex, period and education level among the population aged 25–64 years are shown in Table 1. Among women, all analyzed measures were greater at higher rather than lower education levels. However, in men this only happened for drinking prevalence, prevalence of daily alcohol intake of ≤20 g, HED prevalence and prevalence of 1–3 HED days, since for prevalences of daily alcohol intake >20 g, mean daily drinking amount, prevalence of ≥4 HED days, and mean HED days, the opposite happened. The differences between educational levels were in the same direction in 1997–2007 and 2009–2017 for any drinking measure.

Regarding drinking amount measures, the minimum prevalence of small drinking amounts (≤20 g alcohol/day in men and ≤10 g/day in women) in both periods was observed among the lowest-educated women and the maximum prevalence among the highest-educated men, while the minimum HAD prevalence was observed among the lowest-educated women in 2009–2017, and the highest was observed among the lowest-educated men in 1997–2007. Regarding HED, the minimum values of any measure were observed among the lowest-educated women in 1997–2007 and the maximum values among the lowest-educated men in 1997–2007 (prevalence of ≥4 HED days and mean HED days) or the highest-educated men in 2009–2017 (HED prevalence and prevalence of one HED day) (Table 1). Other general characteristics of the participants by sex, period and educational level are shown in Appendix A.

### 3.2. Educational Inequality in Drinking Amount

The age-standardized measures of drinking amount by sex and education level areplotted in Figure 1. Among men, drinking prevalence (any alcohol amount) and prevalence of intake of ≤20 g of alcohol/day increased significantly with education level (positive gradient). However, prevalences of intake of 21–40 g/day and >40 g/day and mean daily drinking amount decreased significantly with education level (inverse educational gradient). Among women, all measures of drinking amount increased significantly with education level (positive gradient).

The adjusted indicators of educational inequality in drinking amount measures from multivariate regression models are shown in Table 2. After adjusting for age, calendar year, region, marital status and immigration status (model I), the educational gradient among men was significantly positive for drinking prevalence (PC_ye_: 1.3%) and prevalence of daily intake of ≤20 g/day (PC_ye_: 2.9%), while it was significantly negative for prevalence of daily intake of 21–40 g/day (PC_ye_: −1.1%), HAD prevalence (PC_ye_: −6.0%) and mean drinking amount (PC_ye_: −1.8%) (Table 2). Disaggregating HAD prevalence, the PC_ye_ was −5.0% (95% CI: −6.0 to −4.0) for prevalence of 41–60 g/day and −7.3% (95% CI: −8.5 to −6.2) for that of >60 g/day (data not shown in tables). After additionally adjusting for HED frequency (model II), the negative PC_ye_ in HAD prevalence and mean daily drinking amount were attenuated from −6.0% to −4.6% and from −1.8% to −0.8%, respectively, although for HAD prevalence, the change did not reach statistical significance.

Among women the educational gradient in all drinking amount measures was significantly positive, with a PC_ye_ ranging from 3.6% to 5.0%. After additionally adjusting for HED frequency (model II), the gradients in prevalence of intake of 11–20 g/day and HAD prevalence were slightly attenuated, especially the second whose PC_ye_ went from 4.7% in model I to 3.6% in model II, although the changes did not reach statistical significance. The rest of the measures changed little. Similar results were observed by focusing on ratios of each education level to university education (Table 2). The positive educational gradients among the population of women practically disappear when only drinkers in the last 30 days (current drinkers) were considered. In contrast, among current men drinkers, the positive gradient of prevalence of intake of ≤10 g/day attenuated, while the negative gradient of other drinking amount measures became steeper, Appendix A).

### 3.3. Educational Inequality in Frequency of Heavy Episodic Drinking

The age-standardized measures of HED frequency by sex and education level are plotted in Figure 2. Among men, prevalence of having 1–3 HED days in last 30 days increased significantly with education level (positive gradient), while the rest of the HED measures decreased significantly with education level (inverse gradient). Among women, all HED measures increased significantly with education level (positive gradient).

The adjusted indicators of educational inequality in HED frequency measures during the last 30 days from multivariate regression models are shown in Table 3. After adjusting for age, calendar year, region, marital status and immigration status (model I), the educational gradient among men was significantly positive for prevalence of one HED day (PC_ye_: 2.3%) and positive without reaching statistical significance for prevalence of 2–3 HED days, while it was significantly negative for prevalence of ≥4 HED days (PC_ye_: −3.2%), mean HED days (PC_ye_: −4.0%) and also HED prevalence (any HED episode) (PC_ye_: −0.6%) (Table 3). Considering other number ranges of HED days, the PC_ye_ was 1.7% (95% CI: 0.6 to 2.7) for prevalence of two HED days, −2.0% (95% CI: −3.6 to −0.4) for that of three HED days, 1.4% (95% CI: 0.8 to 2.0) for that of 1–3 HED days, −1.4% (95% CI: −2.3 to −0.5) for that of 4–7 HED days and −5.3% (95% CI: −6.4 to −4.2) for that of ≥8 HED days (data not shown in tables). After additionally adjusting for drinking amount (model II), all educational gradients were strongly attenuated. Thus, for prevalence of ≥4 HED days, the PC_ye_ went from −3.2% in model I to −0.7% in model II. Among women, the educational gradient in all HED measures was significantly positive, with a PC_ye_ ranging from 4.6% to 6.0%. After additionally adjusting for drinking amount (model II), the PC_ye_ in prevalence of ≥4 HED days went from 5.3% in model I to 0.9% in model II, while the PC_ye_ in mean HED days went from 4.6% to 0.4%. Similar results were observed by focusing on ratios of each education level to university education (Table 3).

The positive educational gradients among population women practically disappear when only drinkers in the last 30 days (current drinkers) were considered. In contrast, among current men drinkers, the positive gradient in prevalence of one HED day attenuated and almost disappeared, while the negative gradient in prevalence of ≥4 HED days and mean HED days became steeper (Appendix A).

### 3.4. Temporal Evolution of Educational Gradients in Drinking Measures

The direction of the educational gradients was consistent in the periods 1997–2007 and 2009–2017 for all drinking measures. However, there are some between-period differences in the magnitude of such gradients. Among women, the positive educational gradients seem steeper in most recent period for all drinking measures, especially for HED measures, although the differences did not reach statistical significance in any case. However, among men, the changes in the educational gradients for the different drinking measures were not consistent, although the magnitude of the negative gradients in high-risk drinking behaviors as prevalence of HAD or ≥4 HED days increased slightly (Table 4). The important increase in the positive gradients in HED measures among women is explained by the fact that such measures increased more among the highest-educated (those with ≥ secondary 2nd stage education) than the lowest-educated women, although again the differences did not reach statistical significance (data not shown in tables).

### 3.5. Using Other Indicators of Socioeconomic Position

The results on income and occupational inequality in the main drinking measures from model I are shown in Appendix A. Regarding income inequality from 2007–2017, among women, the gradient was significantly positive in all drinking measures, with PCs per thousand euros, ranging from 1.2% to 1.9%, while among men, it was significantly positive in drinking prevalence (0.5%), mean drinking amount (0.2%) and HED prevalence (0.4%), significantly negative in HAD prevalence (−0.6%) and mean HED days (−0.4%), and close to zero in prevalence of ≥4 HED days. Similar results were observed by focusing on ratios of each education level to university education (Appendix A). Regarding occupational inequality from 1997–2017, among women, the ratio of the drinking measure in each occupational class to that in managers/professionals was significantly lower than one in all drinking measures, and its value decreased by decreasing the occupational class, reaching the minimum in unskilled manual workers (positive gradient). However, among men this only happened in drinking prevalence, while in HAD prevalence, prevalence of ≥4 HED days, and mean HED days, the mentioned ratios were in general significantly greater than one, reaching the maximum in unskilled or skilled workers (inverse gradient) (Appendix A).

Finally, we return to the educational inequalities in drinking measures. When the results from model I were further adjusted for income level and occupational class among people aged 25–64 who were at any time employed during 2007–2017, the educational gradients changed little among men and decreased in magnitude among women, especially in HED measures, although their direction remained the same (Appendix A).

## 4. Discussion

### 4.1. The Educational Gradient in Alcohol Intake Strongly Depends on Sex and Measures Analyzed

The results support the first hypothesis that in the Spanish population aged 25–64 years from 1997–2017, the educational gradient in drinking measures varied widely according to sex, especially among men, according to the measures analyzed. As expected, after adjusting for sociodemographic factors, the educational gradient among men was positive (increasing drinking measure with increasing education level) for low drinking amounts and low HED frequencies, but it inverted after a certain threshold (21 g/day of alcohol intake and 3 HED days/month), becoming progressively steeper with increasing drinking amount and HED frequency. Thus, the relative percent change per year of education (PC_ye_) went from −1.1% for a prevalence of 21–40 g alcohol/day to −7.3% for that of >60 g/day, and from −2.0% for a prevalence of two HED days to −5.3% for that of ≥8 HED days. Regarding women, the results only partially support the hypothesis, showing a clearly positive educational gradient for all drinking measures, without decreasing its magnitude when drinking amount or HED frequency increased. In fact, there was little between-measure heterogeneity in the PC_ye_, which ranged from 3.6% to 5.0% for drinking amount measures and from 4.6% to 6.0% for HED measures.

Previous research on directly alcohol-attributable mortality that focused on the Spanish adult population during 2004–2011 found an inverse educational gradient in both men and women with rate ratios of primary to tertiary education of ≥3.8 and ≥1.9, respectively [11]. The direction of these ratios is consistent with the ratios of primary to university education found in our research for HAD prevalence among men (2.2), but not among women (0.6). The same happens with the ratios for prevalence of ≥4 HED days, which were 1.5 among men and 0.6 among women. The educational gradients in high-risk drinking behaviors among Spanish men seem consistent with those found in most previous research in all countries, while those found among women are more similar to those in low- and middle-income countries [13,14,20,25,26,27,28,29]. In any case, the findings show that the direction and magnitude of the educational gradient depends a lot on the analyzed drinking measure, something that has been mentioned previously by some authors [19,26,33]. In addition, this research adds to the literature findings derived from a very detailed analysis of the inequalities in the different intervals of drinking amount and HED frequency, on which there are few published results. Specifically, it has been possible to identify the thresholds from which the positive educational gradient is inverted among men and show that among Spanish women, unlike men, the gradient in drinking measures hardly varies with an increase in the drinking amount or the HED frequency. Likewise, it has made it possible to show that in Spain, the positive gradients in drinking measures among women practically disappear when gradients are calculated among current drinkers. The latter again highlights the heterogeneity of the socioeconomic inequalities in drinking measures according to the specific measure analyzed. The disappearance of the advantage in HAD or HED prevalences in lower- compared to higher-educated women when focusing on current drinkers means that such advantage is due to a lower proportion of drinkers in lower-educated compared to higher-educated women and not to the fact that lower-educated women drinkers have lower alcohol intake or lower HED frequency. Knowing this detail may be interesting in order to better target preventive interventions, but when it comes to explaining socioeconomic inequalities in alcohol-related harm, what is truly relevant is studying socioeconomic inequalities in such drinking behaviors among the entire population of women.

### 4.2. Flattening of Educational Gradient in Binge Drinking When Adjusting for Drinking Amount

The results support the hypothesis that in Spain, the educational gradient in drinking amount contributes to explaining an important part of the gradient in HED frequency measures, but not the opposite. Thus, after additionally adjusting for drinking amount, all educational gradients in HED frequency measures were strongly attenuated in both men and women, moving, for example, the PC_ye_ in prevalence of ≥4 HED days from −3.2% to −0.7% among men and from 5.3% to 0.9% among women. In contrast, after additionally adjusting for HED frequency, the educational gradients in drinking amount measures were only slightly attenuated by moving, for example, the PC_ye_ in HAD prevalence from −6.0% to −4.6% among men and from 4.7% to 3.6 % among women. These findings should not be interpreted as a greater influence of average drinking amount than HED frequency on inequalities in alcohol-related harm. Previous research suggests that HED is important in explaining alcohol-related harm [17,18,23,33]. Our findings do not allow us to elucidate this issue; they simply suggest that in Spain during the analyzed period, most of the possible influences of HED on inequalities in alcohol-related harm would be mediated by average drinking amount. Although there may be an independent pathway between HED and inequalities in alcohol-related harm, the HAD-mediated pathway appears to be of far greater importance. It is interesting to note that in a Finnish study referring to 1998, the SEP gradients in hazardous drinking (prevalence of exceeding weekly hazardous intake) among both men and women were more intensely modified after adjusting for frequent HED than in our research [33].

### 4.3. A Persistent Gender Gap in the Educational Gradient of the Highest-Risk Drinking Behaviors

The findings do not support the hypothesis that the gender gap of the educational gradient in the highest-risk drinking behaviors (as prevalence of HAD or ≥4 HED days) has narrowed in recent years. On the contrary, the gap has been persistent over time and even seems to have widened. Thus, from 2009–2017, the inverse educational gradient in such behaviors among men became steeper than in 1997–2007, and the same happened with the positive educational gradient among women, especially for prevalence 4 HED days, although the differences did not reach statistical significance in any case. In Spain, there are no previous data on the evolution of the SEP inequality in the highest-risk drinking behaviors, and the data from abroad show heterogeneous results [14,43,44,45,46,47].

As noted above, there has been an attempt to explain the positive SEP gradient in the highest-risk drinking behavior among women, especially HED and its evolution, by the theory of diffusion of innovations [13,34,35,36]. According to this theory, these drinking behaviors should have already moved in Spain toward women from lower socioeconomic strata, so the magnitude of their positive SEP gradient (higher values in higher SEP) should have decreased in recent years. However, this has not happened, and even the positive gradient seems to have increased; therefore, other explanations must be sought. One of them could come from the stress-coping theory, according to which the higher-educated women would have a greater alcohol use to cope with work and social stress. This, together with their greater alcohol accessibility and affordability due to their greater economic independence and participation in the labor market, would explain the persistent positive educational gradients [13,27,52,53]. The fact that the magnitude of the educational gradients in the highest-risk drinking measures, especially HED, decreased considerably among women after additionally adjusting for income level and occupational class supports the latter interpretation, suggesting that the positive educational gradient is partly explained by the high income or high occupational status of these women. Likewise, the economic crisis of 2008–2011 and the subsequent fiscal adjustment and austerity policies could have contributed to dampening the increase in alcohol intake among low-educated women. Thus, the results of previous research suggest that during the economic crisis, the negative educational gradient in mortality directly attributable to alcohol could decrease more in women than in men [11]. In any case, it is not ruled out that the positive educational gradient in prevalence of HAD and frequent HED among women may be due to a certain extent to a selection or information bias that implies a greater underestimation of such behaviors in lower- compared to higher-educated women.

### 4.4. Strengths and Limitations

This study is based on a long series of surveys that has allowed a very detailed analysis of some aspects of educational gradients in multiple measures of drinking amount and frequency of heavy episodic drinking, which had been little analyzed and reported previously. The survey methodology regarding analyzed drinking measures remained reasonably consistent over time, and the sample size is considerable. Educational inequality indicators were adjusted for various sociodemographic factors, including age, year, living region, marital status and immigration status. Furthermore, the findings were consistent when other SEP inequality indicators such as income level or occupational class were used. However, the study has some limitations, mainly because self-reported population-based surveys greatly underestimate alcohol use, especially that from heavy drinking, and such underestimation is growing [49,54,55]. The underestimation extent by sex and education is unknown. As noted above, the positive educational gradient in harmful drinking among women could be due in part to a selection or information bias leading to a greater underestimation of such behavior in lower- rather than higher-educated women. Lower-SEP heavy drinkers, especially women, might be more likely to be excluded from the sampling frame and also to reject participation in the survey or be absent from home (bear in mind that EDADES surveys have a general response rate of around 50% and that the substitutes of the non-respondents could have a lower probability of being heavy drinkers). Some research suggests that the out-of-frame and non-responding population have a greater probability of harmful alcohol use [56,57,58,59] and that in general, the probability of non-response is greater among low- SEP people [56,58,60]. Finally, low-educated women might be more likely to hide part of their alcohol intake due to feelings of guilt or shame [61]. On the other hand, this is a cross-sectional design; causality cannot be inferred for the relationship between education level and drinking measures, although recent findings from a Mendelian randomization design suggest a negative effect of educational attainment on HED frequency and total drinks consumed per drinking day and suggest that that positive in alcohol intake frequency may be causal [62]. Part of the educational gradient in drinking measures could be explained by other covariates not considered in this study, including other health-related behaviors. However, as the education level is a fairly distal factor in the causal chains of drinking behaviors and less prone to reverse causality, it is likely that the educational level determines both alcohol use and other health-related behaviors in the same direction. Finally, the education disparities in drinking measures were assessed using relative measures (ratios), which can sometimes even lead to conclusions for those of the absolute measures (differences).

## 5. Conclusions

This research suggests that to adequately monitor SEP inequalities in alcohol intake, various measures of drinking amount and HED frequency should be routinely obtained among men and women. In order to anticipate future SEP inequalities in alcohol-related harm, it is more relevant to select high-risk drinking behaviors such as prevalence of HAD or high HED frequency among the population, but to guide preventive programs, it may also be convenient to select other measures. Currently in Spain, preventive interventions for alcohol intake should be directed more intensively at low-educated men, but also at high-educated women. It is also possible that some of the more serious negative effects on health in high-educated women have not yet manifested themselves, possibly because the increased prevalence of the highest-risk drinking measures in that group is relatively recent. In any case, the positive educational gradient among women was already very clear in 1997–2007. Future research, using other sources of information, should investigate whether the socioeconomic gradients in drinking measures characterized here in the working-age population are similar in the older population. Furthermore, considering that the broad international debate on the alcohol-harm paradox continues, the magnitude of the disparity between the socioeconomic gradients of drinking measures and those of alcohol-related harm measures (mainly mortality and morbidity) should be estimated among men and women in different periods, including cohort designs with a sufficient sample size.

## Figures and Tables

**Figure 1 ijerph-19-04371-f001:**
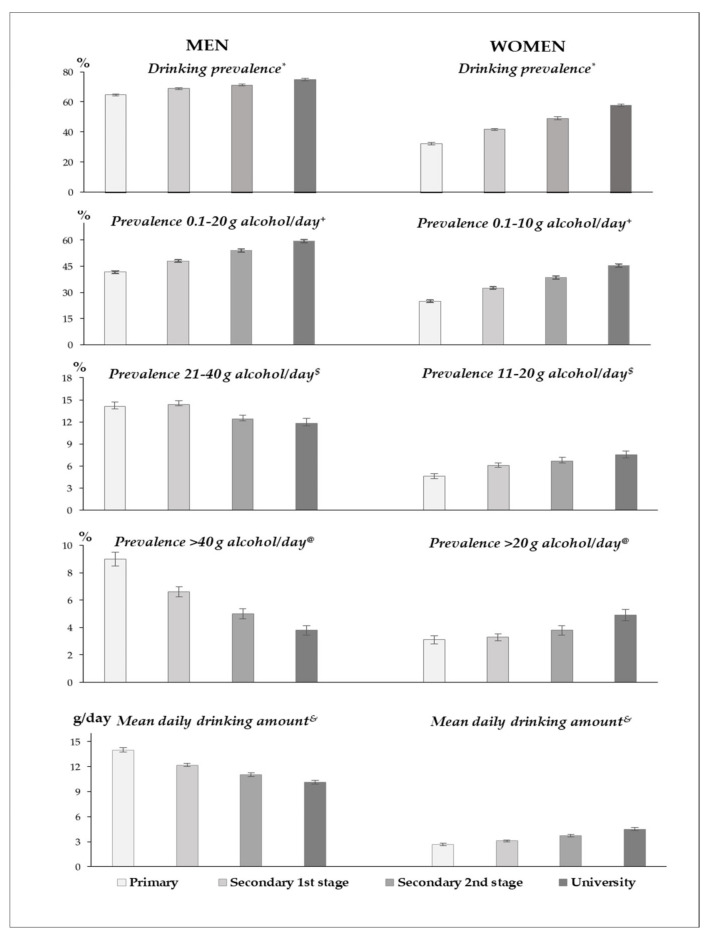
Age-standardized measures of amount of alcohol intake among population aged 25–64 by sex and education level, Spain 1997–2017. **Legend**: Values were directly standardized using the 2013 European Standard Population weights. * Linear trends across education levels were significant both among men (β = 0.05, *p* < 0.001) and women (β = 0.18, *p* < 0.001). **^+^** Linear trends across education levels were significant both among men (β = 0.12, *p* < 0.001) and women (β = 0.19, *p* < 0.001). ^$^ Linear trends across education levels were significant both among men (β = −0.08, *p* < 0.001) and women (β = 0.14, *p* < 0.001). ^@^ Linear trends across education levels were significant both among men (β = −0.31, *p* < 0.001) and women (β = 0.16, *p* < 0.001). ^&^ Linear trends across education levels were significant both among men (β = −0.11, *p* < 0.001) and women (β = 0.17, *p* < 0.001).

**Figure 2 ijerph-19-04371-f002:**
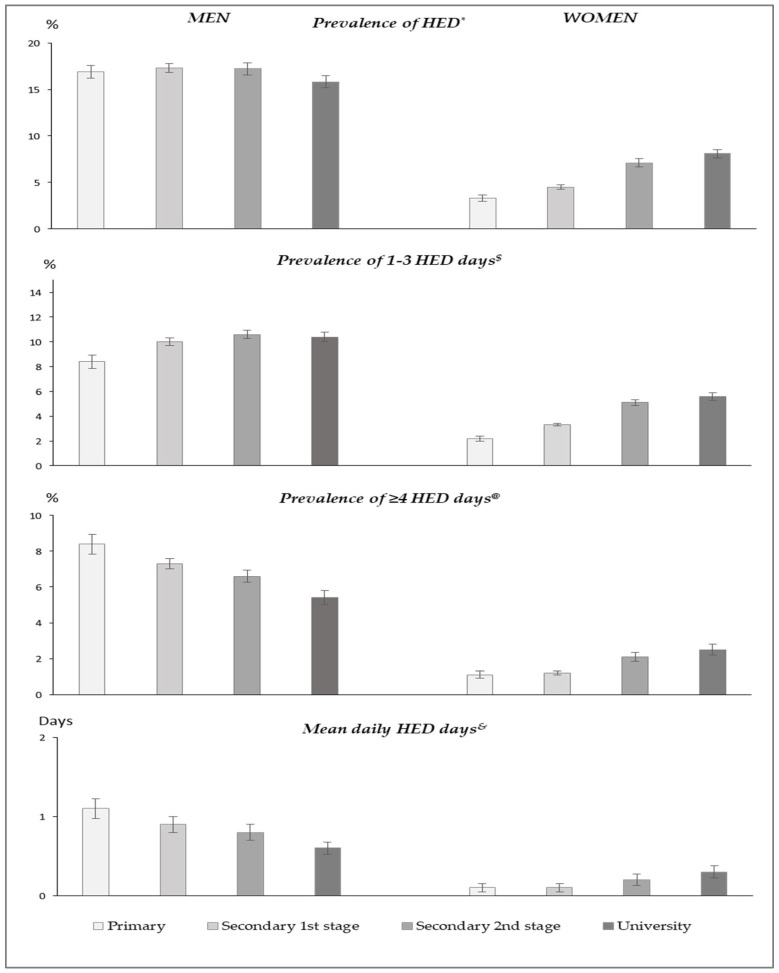
Age-standardized measures of frequency of heavy average drinking among population aged 25–64 by sex and education level, Spain 1997–2017. **Legend**: Values were directly standardized using the 2013 European Standard Population weights. * Linear trends across education levels were significant both among men (β = −0.02, *p* = 0.021) and women (β = 0.32, *p* < 0.001). ^$^ Linear trends across education levels were significant both among men (β = 0.06, *p* < 0.001) and women (β = 0.32, *p* < 0.001). ^@^ Linear trends across education levels were significant both among men (β = −0.14, *p* < 0.001) and women (β = 0.30, *p* < 0.001). ^&^ Linear trends across education levels were significant both among men (β = −0.19, *p* < 0.001) and women (β = 0.29, *p* < 0.001).

**Table 1 ijerph-19-04371-t001:** Crude drinking measures in last 30 days among population aged 25–64 by sex, period and education level, Spain 1997–2017.

**Sex**	** *Men* **			** *Women* **		
**Period**	** * Total * **	** * 1997–2007 * **	** * 2009–2017 * **	** * Total * **	** * 1997–2007 * **	** * 2009–2017 * **
**Measure of alcohol intake and education**						
**Drinking prevalence ^a^ (%)**						
Primary	65.2	67.4	61.8	32.2	32.3	32.2
Secondary 1st stage	69.1	71.7	66.7	42.1	43.9	40.3
Secondary 2nd stage	71.5	74.3	69.1	49.6	51.5	48.1
University	75.3	77.2	73.2	57.8	59.3	56.4
**Prevalence of 0.1–20 g/day in men and 0.1–10 g/day in women ^b^ (%)**						
Primary	41.1	40.5	41.9	24.2	24.0	24.7
Secondary 1st stage	48.6	47.6	49.5	32.9	33.7	32.0
Secondary 2nd stage	55.2	55.5	55.0	39.9	41.1	39.0
University	60.7	60.6	60.8	46.2	46.3	46.2
**Prevalence of 21–40 g/day in men and 11–20 g/day in women ^b^ (%)**						
Primary	15.1	16.1	13.5	5.1	5.0	5.3
Secondary 1st stage	13.8	15.6	12.0	5.9	6.0	5.8
Secondary 2nd stage	11.5	12.7	10.5	6.1	6.1	6.1
University	11.0	12.4	9.6	6.9	7.6	6.4
**Prevalence of HAD ^c^ (%)**						
Primary	9.1	10.8	6.3	2.9	3.3	2.2
Secondary 1st stage	6.7	8.4	5.1	3.4	4.2	2.5
Secondary 2nd stage	4.8	6.1	3.6	3.6	4.3	3.0
University	3.5	4.2	2.7	4.6	5.5	3.8
**Mean daily drinking amount ^d^ (nº grams of alcohol)**						
Primary	14.4	16.2	11.6	2.7	2.9	2.3
Secondary 1st stage	12.2	14.1	10.4	3.1	3.4	2.8
Secondary 2nd stage	10.6	12.2	9.2	3.6	3.9	3.4
University	9.7	10.9	8.5	4.3	4.9	3.8
**Prevalence of HED ^e^ (%)**						
Primary	14.3	13.2	15.9	2.5	1.9	3.6
Secondary 1st stage	18.6	17.2	19.9	4.8	3.5	6.1
Secondary 2nd stage	18.8	17.1	20.4	7.7	5.5	9.4
University	17.5	16.0	19.0	9.4	6.7	11.8
**Prevalence of one HED day ^f^ (%)**						
Primary	3.1	2.4	4.2	0.9	0.6	1.5
Secondary 1st stage	5.3	3.8	6.6	1.9	1.2	2.6
Secondary 2nd stage	6.0	4.0	7.8	3.1	1.9	4.0
University	6.0	4.6	7.6	3.7	2.2	5.1
**Prevalence of 2–3 HED days ^f^ (%)**						
Primary	4.0	3.4	4.8	0.7	0.5	1.0
Secondary 1st stage	5.4	4.5	6.3	1.6	1.1	2.1
Secondary 2nd stage	5.8	5.5	6.0	2.5	2.0	2.9
University	5.7	5.0	6.4	3.0	2.5	3.4
**Prevalence of ≥4 HED days ^f^ (%)**						
Primary	7.2	7.4	6.9	0.8	0.7	1.1
Secondary 1st stage	7.9	8.8	7.1	1.3	1.2	1.4
Secondary 2nd stage	7.0	7.5	6.6	2.1	1.6	2.5
University	5.8	6.5	5.0	2.7	2.1	3.3
**Mean HED days ^g^ (nº days)**						
Primary	1.1	1.1	0.9	0.1	0.1	0.1
Secondary 1st stage	1.0	1.1	0.9	0.1	0.1	0.2
Secondary 2nd stage	0.8	0.8	0.8	0.2	0.2	0.3
University	0.7	0.7	0.6	0.3	0.2	0.3

**Legend: ^a^** Prevalence of any amount of alcohol in the last 30 days **^b^** Prevalence of the indicated daily amount of alcohol intake in grams among men and women **^c^** Prevalence of ≥40 g/day of alcohol intake among men and ≥20 g/day among women. **^d^** Mean daily amount of alcohol intake in grams. **^e^** Prevalence of any HED episode in the last 30 days. **^f^** Prevalence of the indicated number of HED days. **^g^** Mean number of HED days in the last 30 days.

**Table 2 ijerph-19-04371-t002:** Educational inequality in five measures of drinking amount in the last 30 days among people aged 25–64 by sex, Spain 1997–2017.

	*Men*						*Women*					
Measure of Amount of Alcohol Intake and Education	*Model I ^a^*			*Model II ^b^*			*Model I ^a^*			*Model II ^b^*		
**Drinking prevalence ^c^**	*PR ^d^*	*95% CI*		*PR ^d^*	*95% CI*		*PR ^d^*	*95% CI*		*PR ^d^*	*95% CI*	
Primary	0.86	0.84	0.87	0.85	0.83	0.86	0.57	0.56	0.59	0.58	0.56	0.60
Secondary 1st stage	0.91	0.90	0.92	0.89	0.88	0.91	0.74	0.72	0.75	0.75	0.73	0.76
Secondary 2nd stage	0.96	0.94	0.97	0.95	0.94	0.97	0.87	0.85	0.89	0.88	0.86	0.90
University	1.00			1.00			1.00			1.00		
	*PC ^e^*	*95% CI*		*PC ^e^*	*95% CI*		*PC ^e^*	*95% CI*		*PC ^e^*	*95% CI*	
Per year of education	1.33	1.20	1.46	1.46	1.33	1.60	4.82	4.59	5.06	4.70	4.46	4.94
**Prevalence of 0.1–20 g/day in men and 0.1–10 g/day in women ^f^**	*PR ^d^*	*95% CI*		*PR ^d^*	*95% CI*		*PR ^d^*	*95% CI*		*PR ^d^*	*95% CI*	
Primary	0.72	0.70	0.73	0.72	0.70	0.74	0.56	0.54	0.58	0.56	0.54	0.59
Secondary 1st stage	0.80	0.78	0.81	0.81	0.79	0.82	0.72	0.71	0.74	0.73	0.71	0.75
Secondary 2nd stage	0.91	0.89	0.93	0.92	0.90	0.94	0.87	0.85	0.90	0.88	0.85	0.90
University	1.00			1.00			1.00			1.00		
	*PC ^e^*	*95% CI*		*PC ^e^*	*95% CI*		*PC ^e^*	*95% CI*		*PC ^e^*	*95% CI*	
Per year of education	2.95	2.75	3.15	2.80	2.60	3.00	5.02	4.73	5.31	4.97	4.68	5.27
**Prevalence of 21–40 g/day in men and 11–20 g/day in women ^f^**	*PR ^d^*	*95% CI*		*PR ^d^*	*95% CI*		*PR ^d^*	*95% CI*		*PR ^d^*	*95% CI*	
Primary	1.15	1.07	1.23	1.13	1.06	1.21	0.65	0.59	0.72	0.69	0.62	0.77
Secondary 1st stage	1.20	1.13	1.28	1.17	1.10	1.24	0.84	0.77	0.91	0.87	0.79	0.94
Secondary 2nd stage	1.08	1.01	1.15	1.03	0.97	1.10	0.91	0.83	0.99	0.91	0.83	0.99
University	1.00			1.00			1.00			1.00		
	*PC ^e^*	*95% CI*		*PC ^e^*	*95% CI*		*PC ^e^*	*95% CI*		*PC ^e^*	*95% CI*	
Per year of education	−1.12	−1.61	−0.62	−1.10	−1.60	−0.60	3.64	2.81	4.47	3.16	2.31	4.01
**Prevalence of HAD ^g^**	*PR ^d^*	*95% CI*		*PR ^d^*	*95% CI*		*PR ^d^*	*95% CI*		*PR ^d^*	*95% CI*	
Primary	2.19	1.96	2.45	1.78	1.59	1.98	0.56	0.49	0.64	0.63	0.55	0.73
Secondary 1st stage	1.86	1.67	2.06	1.53	1.38	1.69	0.71	0.64	0.79	0.82	0.74	0.92
Secondary 2nd stage	1.40	1.25	1.56	1.25	1.12	1.40	0.81	0.72	0.91	0.85	0.76	0.95
University	1.00			1.00			1.00			1.00		
	*PC ^e^*	*95% CI*		*PC ^e^*	*95% CI*		*PC ^e^*	*95% CI*		*PC ^e^*	*95% CI*	
Per year of education	−5.99	−6.71	−5.25	−4.57	−5.32	−3.81	4.71	3.58	5.86	3.56	2.46	4.68
**Mean daily drinking amount ^h^**	*MR ^i^*	*95% CI*		*MR ^i^*	*95% CI*		*MR ^i^*	*95% CI*		*MR ^i^*	*95% CI*	
Primary	1.24	1.20	1.29	1.11	1.06	1.15	0.57	0.53	0.61	0.57	0.54	0.61
Secondary 1st stage	1.19	1.15	1.23	1.09	1.05	1.13	0.71	0.68	0.75	0.74	0.70	0.77
Secondary 2nd stage	1.10	1.06	1.14	1.05	1.01	1.09	0.85	0.80	0.89	0.85	0.81	0.89
University	1.00			1.00			1.00			1.00		
	*PC ^e^*	*95% CI*		*PC ^e^*	*95% CI*		*PC ^e^*	*95% CI*		*PC ^e^*	*95% CI*	
Per year of education	−1.81	−2.11	−1.51	−0.84	−1.14	−0.54	4.75	4.20	5.30	4.85	4.31	5.38

**Legend**: **95% CI**: 95% confidence interval. ^a^ Negative binomial regression model adjusted for simple age, simple calendar year, region, marital status and immigration status, all entered as dummy variables. ^b^ Same as Model 1, but additionally adjusted for number of days of heavy episodic drinking during the last 30 days. ^c^ Prevalence of any amount of alcohol. ^d^ **PR**: Ratio between the value of drinking measure at each education level and its value at university level. ^e^ **PC**: Relative percent change in the corresponding drinking measure per year of education, under the assumption of a linear association. ^f^ Prevalence of the indicated daily amount of alcohol intake in grams among men and women ^g^ Prevalence of >40 g/day of alcohol intake among men and >20 g/day among women. ^h^ Mean daily amount of alcohol intake in grams. ^i^ MR: Ratio between the value of mean daily drinking amount at each education level and its value at university level.

**Table 3 ijerph-19-04371-t003:** Educational inequality in five measures of frequency of heavy episodic drinking (HED) in the last 30 days among people aged 25–64 by sex, Spain 1997–2017.

	*Men*						*Women*					
Measure of HED frequency and education	*Model I ^a^*			*Model II ^b^*			*Model I ^a^*			*Model II ^b^*		
**Prevalence of HED ^c^**	*PR ^d^*	*95% CI*		*PR ^d^*	*95% CI*		*PR ^d^*	*95% CI*		*PR ^d^*	*95% CI*	
Primary	1.07	1.01	1.13	0.99	0.93	1.04	0.50	0.44	0.56	0.83	0.74	0.93
Secondary 1st stage	1.10	1.05	1.15	1.04	0.99	1.09	0.67	0.61	0.72	0.91	0.84	0.99
Secondary 2nd stage	1.08	1.02	1.13	1.04	0.99	1.09	0.88	0.82	0.95	1.02	0.95	1.09
University	1.00			1.00			1.00			1.00		
	*PC ^e^*	*95% CI*		*PC ^e^*	*95% CI*		*PC ^e^*	*95% CI*		*PC ^e^*	*95% CI*	
Per year of education	−0.56	−0.99	−0.13	0.09	−0.33	0.52	5.60	4.70	6.51	1.21	0.41	2.03
**Prevalence of one HED day ^f^**	*PR ^d^*	*95% CI*		*PR ^d^*	*95% CI*		*PR ^d^*	*95% CI*		*PR ^d^*	*95% CI*	
Primary	0.75	0.66	0.84	0.87	0.77	0.98	0.47	0.38	0.57	0.77	0.63	0.94
Secondary 1st stage	0.90	0.83	0.99	0.98	0.90	1.07	0.65	0.57	0.74	0.87	0.77	1.00
Secondary 2nd stage	0.98	0.89	1.07	1.02	0.93	1.12	0.88	0.77	0.99	0.98	0.87	1.11
University	1.00			1.00			1.00			1.00		
	*PC ^e^*	*95% CI*		*PC ^e^*	*95% CI*		*PC ^e^*	*95% CI*		*PC ^e^*	*95% CI*	
Per year of education	2.26	1.38	3.15	0.99	0.10	1.88	5.98	4.55	7.44	1.58	0.20	2.97
**Prevalence of 2–3 HED days ^f^**	*PR ^d^*	*95% CI*		*PR ^d^*	*95% CI*		*PR ^d^*	*95% CI*		*PR ^d^*	*95% CI*	
Primary	0.95	0.85	1.07	0.97	0.87	1.08	0.47	0.37	0.59	0.80	0.64	0.99
Secondary 1st stage	0.98	0.90	1.08	0.98	0.98	0.98	0.71	0.62	0.83	1.00	0.86	1.16
Secondary 2nd stage	1.01	0.92	1.11	1.00	1.00	1.00	0.87	0.76	1.01	1.02	0.89	1.17
University	1.00			1.00			1.00			1.00		
	*PC ^e^*	*95% CI*		*PC ^e^*	*95% CI*		*PC ^e^*	*95% CI*		*PC ^e^*	*95% CI*	
Per year of education	0.56	−0.30	1.43	0.42	−0.44	1.28	5.39	3.72	7.08	0.88	−0.62	2.41
**Prevalence of ≥4 HED days ^f^**	*PR ^d^*	*95% CI*		*PR ^d^*	*95% CI*		*PR ^d^*	*95% CI*		*PR ^d^*	*95% CI*	
Primary	1.48	1.34	1.63	1.10	1.00	1.20	0.57	0.46	0.71	0.96	0.77	1.18
Secondary 1st stage	1.40	1.29	1.53	1.15	1.05	1.24	0.63	0.54	0.74	0.89	0.76	1.04
Secondary 2nd stage	1.23	1.13	1.35	1.11	1.02	1.21	0.89	0.77	1.03	1.08	0.93	1.25
University	1.00			1.00			1.00			1.00		
	*PC ^e^*	*95% CI*		*PC ^e^*	*95% CI*		*PC ^e^*	*95% CI*		*PC ^e^*	*95% CI*	
Per year of education	−3.16	−3.85	−2.46	−0.75	−1.45	−0.06	5.29	3.54	7.07	0.89	−0.64	2.44
**Mean HED days ^g^**	*MR ^h^*	*95% CI*		*MR ^h^*	*95% CI*		*MR ^h^*	*95% CI*		*MR ^h^*	*95% CI*	
Primary	1.62	1.47	1.80	0.99	0.90	1.09	0.61	0.50	0.74	1.01	0.86	1.19
Secondary 1st stage	1.47	1.35	1.60	1.11	1.02	1.20	0.68	0.60	0.77	0.94	0.84	1.05
Secondary 2nd stage	1.23	1.13	1.35	1.09	1.01	1.19	0.92	0.82	1.04	1.15	1.03	1.29
University	1.00			1.00			1.00			1.00		
	*PC ^e^*	*95% CI*		*PC ^e^*	*95% CI*		*PC ^e^*	*95% CI*		*PC ^e^ *	*95% CI*	
Per year of education	−3.97	−4.73	−3.19	−0.06	−0.77	0.64	4.56	3.15	5.99	0.38	−0.76	1.53

**Legend**: **95% CI**: 95% confidence interval. **^a^** Negative binomial regression model adjusted for simple age, simple calendar year, region, marital status and immigration status, all entered as dummy variables. **^b^** Same as Model 1, but additionally adjusted for number of grams/day of alcohol intake during the last 30 days. **^c^** Prevalence of any HED episode in the last 30 days. **^d^ PR**: Ratio between the value of the HED measure at each education level and its value at university level. **^e^ PC**: Relative percent change in the corresponding HED measure per year of education, under the assumption of a linear association. **^f^** Prevalence of the indicated number of HED days in the last 30 days. **^g^** Mean number of HED days in the last 30 days. **^h^ MR**: Ratio between the value of mean HED days at each education level and its value at university level.

**Table 4 ijerph-19-04371-t004:** Educational gradients in drinking amount and frequency of heavy episodic drinking in the last 30 days among population aged 25–64 by sex and calendar period, Spain 1997–2017.

	*Men*						*Women*					
	*1997–2007*			*2009–2017*			*1997–2007*			*2009–2017*		
**Drinking amount measure**	*PC ^a^*	*95% CI*		*PC ^a^*	*95% CI*		*PC ^a^*	*95% CI*		*PC ^a^*	*95% CI*	
Drinking prevalence ^b^	1.2	1.0	1.4	1.6	1.4	1.8	4.6	4.3	4.9	5.0	4.6	5.3
Prevalence of HAD ^c^	−6.0	−6.8	−5.1	−6.4	−7.7	−5.0	4.2	2.7	5.6	5.4	3.5	7.3
Mean daily drinking amount ^d^	−2.1	−2.5	−1.7	−1.5	−2.0	−1.0	4.4	3.7	5.2	5.2	4.4	5.9
**HED measure**												
Prevalence of HED ^e^	−0.7	−1.3	−0.1	−0.5	−1.1	0.1	4.0	2.6	5.5	6.3	5.2	7.5
Prevalence of ≥4 HED days ^f^	−2.9	−3.8	−2.0	−3.8	−4.9	−2.6	2.6	0.1	5.3	7.0	4.7	9.4
Mean HED days ^g^	−4.3	−5.3	−3.3	−3.8	−4.9	−2.6	2.8	0.8	4.8	5.7	3.8	7.6

**Legend**: **95% CI**: 95% confidence interval. **^a^ PC:** Relative percentage change in the corresponding drinking measure per year of education, under the assumption of a linear association, obtained from negative binomial regression model adjusted for simple age, simple calendar year, region, marital status and immigration status, all entered as dummy variables. **^b^** Prevalence of any amount of alcohol. **^c^** Prevalence of ≥40 g/day of alcohol intake among men and ≥20 g/day among women. **^d^** Mean daily amount of alcohol intake in grams. **^e^** Prevalence of any HED episode in the last 30 days. **^f^** Prevalence of ≥4 HED days in the last 30 days. **^g^** Mean number of HED days in the last 30 days.

## Data Availability

Data sets used and/or analyzed during the present study should be requested for use from DGPNSD.

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
