# Peer review of "Educational Gradients in Drinking Amount and Heavy Episodic Drinking among Working-Age Men and Women in Spain"

_ijerph, 2022, doi:10.3390/ijerph19074371_

Round 1

Reviewer 1 Report

This study aims to investigate the educational gradient in monthly measures of drinking amount and heavy episodic drinking among people aged 25-64
years in Spain. Trends in alcohol intake by education have been extensively studied. However, the authors put an afford to change the direction of their study and added a new valid rationale to their study. I recommend accepting the manuscript after English language editing.

Author Response

The English translation has been thoroughly revised.

The discussion has been improved and completed.

Reviewer 2 Report

Thanks for addressing my comments. I have no further comments.

Author Response

The English translation has been thoroughly revised.

Reviewer 3 Report

The authors made all the requested changes

Author Response

(The authors gave the same response as above.)

Reviewer 4 Report

In our opinion, the study is of great scientific interest and should be published.
However, we consider that a greater discussion of the limitations derived from the participation rate being 50% (effects on prevalence, on associations...) should be included.

Author Response

The discussion has been completed and improved by addressing the limitations arising from the 50% participation rate. In particular, the following excerpt has been added:

“As noted above, the positive educational gradient in harmful drinking among women could be due in part to a selection or information bias leading to a greater underestimation of such behaviour in lower than higher-educated women. Lower-SEP heavy drinkers, especially women, might be more likely to be excluded from the sampling frame and also to reject participation in the survey or be absent from home (bear in mind that EDADES surveys have a general response rate of around 50% and that the substitutes of the non-respondents could have a lower probability of being heavy drinkers). Some research suggests that the out-of-frame and non-responding population have a greater probability of harmful alcohol use [56, 57, 58, 59] and that in general the probability of non-response is greater among low -SEP people [56, 58, 60]. Finally, low-educated women might be more likely to hide part of their alcohol intake due to feelings of guilt or shame [61].”